# Integration of Obese Children in Physical Education Sessions: An Exploratory Study

**DOI:** 10.3390/children10010133

**Published:** 2023-01-10

**Authors:** Bilel Aydi, Okba Selmi, Santo Marsigliante, Mohamed A. Souissi, Nizar Souissi, Antonella Muscella

**Affiliations:** 1Research Unit, Sportive Performance and Physical Rehabilitation, High Institute of Sports and Physical Education of Kef, University of Jendouba, EI Kef 7100, Tunisia; 2Research Unit, Sportive Sciences, Health and Movement, UR22JS01, El Kef 7100, Tunisia; 3High Institute of Sport and Physical Education of Sfax, University of Sfax, Sfax 3000, Tunisia; 4Department of Biological and Environmental Science and Technologies (DiSTeBA), University of Salento, 73100 Lecce, Italy; 5Physical Activity, Sport, and Health, UR18JS01, National Observatory of Sport, Tunis 1003, Tunisia; 6High Institute of Sport and Physical Education, Ksar-Saïd, Manouba University, Tunis 1003, Tunisia

**Keywords:** children with obesity, psychological response, training load, well-being enjoyment, mood, motivation, small-sided games

## Abstract

We investigated the effect of the role of the joker in children with obesity (OCs) on integration and physio-psychological responses during small-sided games (SSG) training programs. Sixteen OC students (age 13.8 ± 0.73 years) performed training programs consisting of two sessions a week for three weeks. The experimental protocol consisted of 16 teams of 4 children (3 of normal weight and 1 OC). The 16 teams were divided into 2 groups, one with an OC playing as the joker (SSG-J) and the other group with OC playing as non-joker (SSG-NJ). Maximum heart rates (HRmax), blood lactate concentration [La] and OMNI-Child perceived exertion were measured at the end of each SSG. A physical activity enjoyment Scale (PACES) was accomplished during physical activity for the evaluation of feelings in OCs. Additionally, the profile of mood states (POMS) was measured before and after the SSG-J and SSG-NJ programs. HRmax, [La], perceived exertion, and PACES scores were significantly higher after the SSG-J compared with SSG-NJ (increments of 6.4%, 31.7%, 19.5% and 18.1%, respectively). The score of the POMS variables was positively increased in the presence of jokers. The vigor score increased by 30%, while tension and total mood disturbance scores decreased by 27.6% and 4.5%, respectively. These findings suggest that the joker role could be effective in improving integration, physical enjoyment, physiological responses and mood states in OCs when a team game is used during PE sessions. PE teachers could then program joker exercises with the aim of improving OCs’ physical commitment willingness to play.

## 1. Introduction

Obesity in children has become a major health problem all over the world. Recent estimates from 2020 showed that between the ages of 5 and 19, there are over 340 million children and adolescents who are overweight or obese [1]. These states are associated with a raised risk of physical and physiological diseases as well as serious psychological consequences [2]. Thus, childhood obesity is a challenge for health systems worldwide, and many international organizations are putting strategies and plans in place to decrease its prevalence [3]. Today’s children lead an increasingly sedentary lifestyle that involves time spent playing video games, using computers/smartphones and watching television [4]. This lifestyle leads them to neglect the physical activity (PA) typical of this period of development [5]; this has negative implications in that PA has several positive effects on the growth of children and teens during puberty.

In September 2020, the WHO Regional Committee for Europe proposed a new European Strategy of Work for the period 2020–2025 that focuses on united actions for improving health by promoting physical activity and healthy nutrition to contrast obesity [6]. PA in children and adolescents benefits a range of medical conditions, including cardiovascular disease, obesity and a range of psychological health problems [7]. However, usually, children with obesity (OCs) find a lot of difficulty in PA sessions [8]. In fact, research indicates that obesity leads to fatigue, tension, low self-esteem, poor body image and exercise avoidance [9]. In addition, in school, OCs are often stigmatized as being unmotivated, less competent, lacking in self-discipline or not athletic enough to participate in physical education (PE) [8,10]. It is well documented that teasing about obesity is linked to psychological and emotional problems in children which make them dissatisfied with their PE sessions [11]. During PE sessions, exercises can lead to greater energy demands on OCs than on non-obese pupils. For this reason, they find it difficult to fit in physically and affectively, and to communicate with others. 

Effective strategies for the integration and encouragement of OCs require the participation of the school environment [12]. OCs generally have lower motor skills and physical condition than normal-weight children; this means it is an extra challenge for PE teachers to include them in PE sessions [13,14]. It is necessary for PE teachers to provide individualized and differentiated exercises for OC to integrate them positively into the PE lessons. The PE teacher must create inclusive and safe learning climates for OC in PE classes or provide different situations (i.e., methods, exercises) in terms of their unique characteristics [15]. 

The role of the PE teacher in the motivation and integration of OCs in the group can contribute effectively to the development of more positive physical, behavioral and emotional states, which gives them more confidence and integration in the group [16,17]. 

Mostly, the need to differentiate teaching obliges PE teachers to choose methods and styles of learning to cater to the needs of schoolchildren [18]. One of the more definite pedagogical models in the field of PE to introduce students to sports training is the one specified as Teaching Games for Understanding (TGfU). The TGfU pedagogical approach improves decision making, enjoyment, psychological aspects and students’ intention to be physically active [19,20]. 

Some studies indicated the effectiveness of TGfU in increasing OCs’ motivation, engagement, enjoyment and cognitive learning [21,22]. Generally, the programming of specific and adapted exercises for OCs in PE sessions resulted in positive changes in physical engagement and student motivation, with positive effects on affective aspects [13,23]. Physical activities help children and adolescents to decrease anxiety, improve mood and progress their mental health [24,25]; thus, stimulant PE sessions could be an effective strategy to integrate OC and promote positive physical engagement and communication with other students [13]. In addition, motivation and enjoyment achievement during PE is of wider interest for the correlation between enjoyment, intrinsic motivation and being physically active [26,27].

In addition, motivational exercises in PE sessions have been shown to modulate the emotional and physiological states of students, which may contribute to its beneficial effect on the intensity of effort, positive feeling and higher physical enjoyment [28]. On the other hand, non-motivating running exercises negatively affect mood state and pleasure in OCs [29,30]. The role of the PE teacher in the motivation and integration of OCs in the group effectively contributes to the development of more positive physical, behavioral, and emotional states in these OCs, which give them greater confidence and integration into the group [16,17].

As the motivating exercises are fun for the students, they can encourage OCs to become more active [31].

In support of this, enjoyment has demonstrated links to motivation and PE teacher encouragement [32]. Research indicates that in PE sessions, collective game exercises such as small-sided games (SSG) are beneficial in terms of motivation, integration and communication between students [33]. SSGs are widely used as a methodological strategy in games teaching, depending on teacher’s aim. For example, Pinho et al. [34] indicated that recreational SSG programs were able to improve maximal physical fitness and maintain psychological balance in overweight and OCs. Lofrano-Prado et al. [35] reported that exercising at high intensity leads to negative feelings and decreased well-being, decreased vigor, increased fatigue and tension and altered mood states in adolescent obese people. 

These types of exercises are favored by students because they find motivation, commitment and opposition [36,37]. Recently, it has been proposed that SSGs are also used for affective solicitation. In this context, research have indicated that this exercises type is more effective pedagogically than other conventional exercises in producing a positive mood state and greater physical enjoyment [38,39,40]. 

Thus, PE enjoyment and positive mood states may play a crucial role in physical activity engagement more generally in OCs.

We can claim that the role of the PE teacher and the modality of exercises specific to OCs have an important role in determining the motivation and the positive feelings. As far as we know, no studies have referred to the effects of the integration of OCs in SSG exercises on psychophysiological and affective responses during SSGs in PE sessions.

Thus, the goal of this research was to study the impact of the OC’s position in the role of joker on psychophysiological aspects, physical enjoyment and mood state while completing SSG exercises (games of 10 successive passes) in PE sessions. The joker is used to assist other players because it is involved in both the defensive and offensive phases of the game. More, the role of joker player constrains the game by forcing the defender player to adapt to a new game context [41]. Using joker players is a common practice during training sessions [41,42,43]; however, there are no investigations regarding OC integration.

Using the joker position during ball games in PE sessions could be important and beneficial for OC, because it is more effective for learning and motivating than other methods. Furthermore, this specific intervention could provide physiological and psychological benefits due to positive sensations and high effort.

## 2. Materials and Methods

### 2.1. Participants

Sixteen children with obesity (OC) belonging to the 7th and 8th basic education participated in the study; they practiced two physical education sessions per week for a total duration of 3 h. The characteristics of the OC are presented in Table 1.

The study was conducted during mandatory physical education lessons. The inclusion criteria were: (1) all OCs competed for the same school; (2) all OCs belong to same physical education teacher (3) no illnesses or injuries experienced during the study and 2 months prior to the study; (4) no previous cognitive or physical disease (5) the body mass index (BMI) of OCs was greater than 30 kg·m^–2^. These were the exclusion criteria: (1) no regular presence of OCs in PE lessons; (2) OC became sick during the trial period. Children and parents voluntarily joined this research, and after a detailed explanation of the objectives and risks involved in this research, they gave their consent. 

We performed a sample size calculation using G*Power software. The analysis confirmed that 16 subjects were sufficient to detect significant differences with an 84.87% chance of correctly rejecting the null hypothesis.

The study was conducted according to the Declaration of Helsinki guidelines, and all procedures were approved by the local ethics committee of the High Institute of Sports and Physical Education of Kef, Jendouba, Tunisia (approval no. 012/2021).

### 2.2. Anthropometric Assessments

Measurement of children’s’ height and body mass were conducted when the subjects were without shoes, barefooted and only wearing light shorts. The height of the children was recorded in centimeters with an accuracy of 0.1 cm using a Harpenden stadiometer. During height measurement, the obese children were in an upright position, with their heels together, arms extended, and head positioned parallel to the floor. The body mass of the children was recorded in kilograms (measured to the nearest 0.1 kg) using an electronic scale (Tanita, Model TBF-410 GS, Tokyo, Japan). During body mass measurement, OCs stood upright at the center of the platform of the balance with their arms extended. Body mass index (BMI) was calculated as weight/height squared (kg·m^–2^) (BMI = weight (kg)/height (m)^2^).

### 2.3. Procedure

This study analyzed the psychological and physiological responses, mood state and enjoyment of schoolchildren during SSG using hands (games of 10 successive passes) according to two conditions (SSG with joker and SSG without joker). The use of joker players promotes the rotation on both game phases (offense and defense). The changing environment constrains players to explore more task solutions and consequently enlarges the exploratory breadth of their tactical behavior. The research was conducted during the 2021–2022 scholar midseason (16 weeks after the beginning of the season). Before the experimental sessions, the anthropometric characteristics were measured and the maximum heart rate (FCmax) of the OCs was estimated using the VAMEVAL test.

During regular PE lessons, 2 sessions of a 4-a-side SSG were fulfilled, split by a one-week interval. Each testing protocol was repeated once and consisted of so-called SSG-J teams (3 vs. 3 normal weight children plus 2 OCs participating as jokers in each team) and SSG-NJ teams (3 vs. 3 normal weight children plus 2 non-joker OCs in each team); therefore, there were a total of 16 teams and each team was composed of 3 normal weight children and 1 OC. During the experimental week, entrants were divided into 2 groups with 8 OCs completing SSG-J and the other 8 completing SSG-NJ in a randomized and counterbalanced order. In total, each OC completed the SSG-J and the SSG-NJ once (Figure 1). The length of each SSG intervention was 18 min. The profile of mood states (POMS) was evaluated before and after each trial (SSG-J and SSG-NJ) and the HR was continuously monitored during each session. Furthermore, RPE, a fingertip blood sample and physical activity enjoyment scale (PACES) were recorded immediately after the trial. OCs finished PACES separately so as not to listen the other children’s responses. All measurements were taken on the same school indoor sports hall during regular PE lessons. All OCs refrained from strenuous exercise for at least 48 h prior to experiments. All OCs became accustomed to the RPE, PACES and POMS questionnaire and to the SSG system prior to the start of the experimental session in order to minimize any learning effects during the study. The same PE teacher collected the data obtained in the tests of each session.

### 2.4. Small-Sided Games Session

A standardized 12 min warm-up was given to all students (OCs and normal weight students who participated in the physical education session), consisting of low-intensity running, muscle coordination exercises, dynamic stretching and drills to pass the ball with the hands and ending with 3 × 10 m sprints. During the warm-up, no stretching exercises were performed and the first SSG bout was performed 3 min after. SSG exercises were played with handball balls without goalkeepers.

Both the SSG-J and SSG-NJ format played on a 20 × 10 m pitch (25 m^2^ per player). The trial lasted 18 min (four bouts of 3 min separated by 2 min of passive recovery). The team that makes 10 successive passes gets a point. At the end of each bout, the losing team will be sanctioned by the physical education teacher (10 pumps or 10 abs or 10 squats). The SSG-J group played the game with the participation of 2 OC jokers who played with both teams. It is obligatory that each OC joker pass the ball at least once during each 10 successive passes otherwise the point will not be counted. During the SSG-NJ, the OC children played without the obligation to pass during the 10 successive passes. During the SSG, students were required to keep possession of the ball for as long as possible, to compete with maximum effort and to perform more series of 10 successive passes to obtain a greater number of points. Students are required to take no more than 3 steps with the ball (more than 3 steps is considered walking) and not to return the pass directly to the same student who gave the pass to ensure all participants engaged in the game. The PE teacher moved about the outer edge of the field while encouraging the students using physical education-specific terminology and vocabulary (i.e., more active, ‘go go go’, ‘again again’, ‘seek the ball’, ‘keep the ball’, ‘more’ and so on) and, when needed, providing new balls to keep the exercise going continuously.

### 2.5. The Progressive Field Test for the Evaluation of Maximum Heart Rate (VAMEVAL)

Before the beginning of the experimental sessions, the OCs performed the VAMEVAL test on a 200 m running track to estimate their maximum heart rate (HRmax) as previously described [40]. A Polar Team Sport System (Kempele, Finland) was used in order to record HR. HR data were expressed as a percentage of VAMEVAL-HRmax (%HRmax), and mean HR (HRmean). %HRmax for SSG-J and SSG-NJ sessions was calculated by the following formula: %HRmax = (HRmean/VAMEVAL-HRmax) ∗ 100 [44]. The reliability (Cronbach’s α of 0.83) of the VAMEVAL protocol as a tool for assessing HRmax has been previously shown [45].

Three min after SSG-J and SSG-NJ exercises were collected, blood samples from the fingertip in the absence of active recovery and the concentration of blood lactate [La] were assayed by Lactate Pro Analyzer (Arkray, Tokyo, Japan) [46].

### 2.6. The OMNI-Child Perceived Exertion Scales

Immediately, at the end of each SSG, perceived exertion was evaluated using the children’s OMNI-RPE (0–10) scale [47]. The OMNI scale (0 to 10) was used to measure RPE during exercise, with descriptive terms combined with images to help interpret numerical values. For the OMNI-RPE, the children were asked the following question: “How tired does your body feel during exercise?”. Each obese child’s RPE was collected ensuring that perceived effort referred solely to the SSG intervention and was validated in PE [48]. The children responded individually, preventing them from knowing the scores of the other participants. 

### 2.7. The Profile of Mood State

The profile of mood states (POMS) questionnaire [49] was employed in order to assess changes in mood during physical activity. The questionnaire was dispensed to all OCs 15 min before and 5 min after each SSG to measure mood state. The POMS questionnaire was made up of 24 elements projected to evaluate 6 emotional states (tension/anxiety, anger/hostility, fatigue/inertia, depression/dejection, confusion/perplexity, and vigor/activity). The response format is a 5-point rating scale (0 = not at all, 1 = a little, 2 = moderately, 3 = quite a bit, 4 = extremely). The total mood disturbance (TMD) score is obtained by adding the scores of 5 negative moods and subtracting the score of the positive moods; to prevent obtaining negative numbers, a constant value of 100 was then added. 

Cronbach’s α ranged from 0.86 to 0.93 (in detail: 0.88 for anger; 0.91 for confusion; 0.86 for depression; 0.90 for fatigue; 0.88 for tension and 0.93 for vigor). Players completed the POMS individually on papers on the same school indoor sports hall where they completed the SSG sessions.

### 2.8. Physical Enjoyment

The 8-item physical activity enjoyment scale (PACES) was performed in order to assess the positive feelings coupled with physical activity in OC [50]. Five minutes after each SSG exercises, PACES was used to measure enjoyment. The OC were asked to rate “how you feel at the moment regarding the physical activity you experienced” using a 7-point rating scale ranging from 1 = it is very pleasant, to 7 = it is not fun at all. The total physical enjoyment score was evaluated using the sum of the 8 items’ scores which yielded possible ranges from 8 to 56 points. Higher PACES scores indicate higher levels of physical enjoyment. In the current study, the Cronbach’s α value of the PACES test was 0.90. Players answered the questionnaire individually.

### 2.9. Statistical Analysis

SPSS statistical analysis software (20.0 version, SPSS Inc., Chicago, IL, USA) was used for all statistical analyses.

The hypothesis of normality and the homogeneity of the variance were verified using the Kolmogorov–Smirnov test. The Mann–Whitney test was used to examine the differences between the training programs with and without joker players.

To evaluate the effect of “Group” (with and without joker players), “Time” (pre and post training) and “Interaction” (Group × Time) on mood responses (POMS scores), a two-way analysis of variance (ANOVA) was used. The results were considered significant at the 95% confidence level (*p* ≤ 0.0).

## 3. Results

### 3.1. Internal Load Responses

HRmax and RPE values were significantly higher during joker role play (*p* < 0.0001 and *p* < 0.01, by the Mann–Whitney test, respectively; Table 2). 

For [La], an effect of time point was recorded (after each SSG exercise: SSG-J and SSG-NJ) with higher values recorded during the joker role (*p* < 0.01, Table 2).

### 3.2. Physical Enjoyment 

Perceived enjoyment was significantly higher (*p* < 0.0001; ES = 5.35) after the small-sided games with joker training (PACES score = 47.06 ± 3.04) compared with those without joker training (PACES score = 39.81 ± 2.99).

### 3.3. The Profile of Mood State 

The profile of mood states (POMS) was used in order to assess OCs’ distinct mood states. The scores of POMS were assessed before and after SSG-J and SSG-NJ training programs.

Bidirectional analyses of variance with repeated measures (group × time) performed among POMS variables are shown in Table 3. The results showed a significant effect of the joker player and of time on fatigue (group: *p* < 0.001, η^2^ = 0.262; time: *p* < 0.001, η^2^ = 0.309; Table 3).

We observed an effect of group-to-group interaction time on TMD interaction; it decreased after the small-sided games with joker training (*p* < 0.05, by ANOVA), as result of a reduction in tension (*p* < 0.001, by ANOVA) and increase in vigor (*p* < 0.001, by ANOVA) scores (Figure 2). 

## 4. Discussion

Obesity in children has become a major health problem all over the world. Physical inactivity and a lack of sufficient physical activity among young students are factors in the growth of BMI and obesity among children. Thus, the Federal Centre for Health Education aims to improve the health of children and adolescents aged 12–18 years by carrying out an awareness initiative regarding healthy lifestyle that will be promoted in schools and via media and the internet [1]. Nevertheless, the experiences of children with obesity in schools hint that PA may concur with to their marginalization. School-based PE could be awkward for OCs, bringing the overweight body to the fore and rendering children at risk of peer disapproval [51]. During PA, beginning and keeping a positive, inclusive social climate in school is pivotal, particularly given the great number of potential social strains for OCs. The PE teacher should choose a pedagogical approach that enhances decision making and enjoyment and especially improves students’ intention to be physically and psychologically active (motivation, responsibility, fun) [21]. Quite frequently in the game of soccer, various coaches introduce the role of the joker in the SSG method as a training system for players. More precisely, the SSGs improve the player’s decisions and actions. Various movement activities, such as tactical behavior and technical skills and the physiological responses occurring during the game are effortlessly applied by the trainers through adding diverse rules, purposes or ends in SSGs [52]. Thus, in a performance context, players can uncover and understand a wide set of possibilities to act, performing task solutions by adapting their behavior to the actions of their adversaries and teammates [53].

To maximize the functional action versatility [54] and increase the exploratory efficiency of players, football coaches usually introduce joker players during training drills.

Moreover, it was seen that the use of joker players, which gives numerical superiority/inferiority in SSGs, produces dissimilarities in the external and internal load reactions of both the joker and the main players [55].

Therefore, we explored a PE intervention geared towards including students with obesity. Particularly, the purpose of this study was to examine the impact of the joker role of OCs on psychophysiological aspects, physical enjoyment and mood state while completing SSG exercises (games of 10 successive passes) in PE sessions. The main findings of the present study are that (1) SSG-J increased internal intensity, HRmax and [La] more than occured in the SSG-NJ condition, (2) the physical pleasure experienced with SSG-J was greater, and (3) SSG-J produced a far more positive mood than SSG-NJ. 

Few studies have been conducted on SSGs that include joker players, and the researchers obtained different results in these studies; the present study is the first to elaborate on the impact of the joker role during physical activity on the physiological and affective aspects in OCs during PE sessions. 

### 4.1. Internal Load Responses

Since the joker role leads to an increase in the values of RPE, %HRmax and [La] (16.30%, 6.09%, and 24.08%, respectively), OCs performed the SSG-J game with higher intensity, resulting in higher solicitation of cardiorespiratory demands. This may well be due to the increased effort and higher level of physical engagement exerted by the joker player who moves well to touch the ball multiple times.

During the SSG-J, the RPE is harder (>6) and %HRmax is higher (>84%) than in SSG-NJ game, suggesting that the high level of internal intensity and HR are linked to the effort produced by children during exercise training. 

There are few studies carried out on SSG with jokers; moreover, the results obtained are conflicting. For example, the study by Sanchez-Sanchez shows that the lowest RPE responses are observed in games with joker players [42], while that by Hill-Haas et al. [41] indicates that the RPE responses of joker players were higher compared with core players, and that the [La] responses were similar [41]. The results of our study differ from the latter and, therefore, further research is needed with the aim of evaluating the RPE and [La] responses of joker players. As is known, football is played in conditions of quantitative imbalance, and there are very few studies on the use of jokers capable of achieving such numerical inferiority/superiority [56,57]. 

On the other hand, our results confirmed those of Kumak et al. [55] that indicated that the games with 2 jokers (3 vs. 3 + 2j) caused higher values in terms of RPE and HRmax compared with a lack of joker during specific soccer training on a 15 × 25 m field. In addition, another recent study showed that the internal and external load were significantly higher in SSGs with a joker (4 vs. 4 + 2j) than during SSGs without a joker in soccer players [58]. Regarding [La] in our study, PE during SSG-J caused its increase, suggesting the contribution of anaerobic metabolism to energy production [40,59]. Conversely Kumak et al. [55] reported that [La] and the rate of perceived exertion of the main players were significantly higher compared with joker players. 

Playing with jokers could also provide some motivation for the participant; this has significant implications for adherence and physical performance utilizing a practical participation. Such results indicate that the physiological reactions and the internal intensity could vary depending on the motivation of the entrants and the PE teacher’s encouragement during the games [60]. The results of this study are coherent with those described by Sahli et al. [38], who showed that motivation in SSG exercises during PE sessions positively affects physiological aspects (%HRmax and La) and internal intensity among students. Overall, these findings showed that SSG-J provoked greater anaerobic and aerobic contributions to energetic requests and internal intensity, thus endorsing the meaning of the joker role in the progress of OCs’ physical fitness.

### 4.2. Psychological Responses between SSG-J and SSG-NJ

Concerning the comparison of psychological responses between SSG-J and SSG-NJ, many studies emphasized the significance of the POMS questionnaire and PACES scale in assessing affective states in students during PE sessions [33,38,61]. Physical activity with a joker has been positively associated with affective appearance to motivate OCs to improve their mood state and physical enjoyment through participation in the game. The present study indicates that SSG-J produces greater physical enjoyment and positive mood than SSG-NJ. These results demonstrate that positive feeling during a PE session is influenced by the motivation variable [62,63]. As a matter of fact, physical enjoyment represents a positive feeling associated with physical activity [64]. The present study showed that the joker role has a positive effect on physical enjoyment with PACES score increment (15.40%). In this regard, the role of the joker during SSG is linked to positive affective responses to the exercise; this is one of the main reasons why OCs contribute to physical activity. We suggest that motivational factors and the PE teacher’s verbal encouragement explain the positive feeling of physical enjoyment during the SSG-J condition. This result agrees with Navarro-Patón et al. [32] who indicated that motivation during PE sessions is expected to increase students’ physical enjoyment and improve positive behavior.

Similarly, Engels and Freund [65] indicated that motivating exercises during PE sessions are expected to increase students’ enjoyment and improve positive behavior. In this regard, motivation during SSG is associated with positive emotional responses to physical activity and is one of the primary causes that stimulate participants to agree to the game. These results also propose that the physical enjoyment due to physical activity may change reliance on exercise formality, the PE teacher’s verbal encouragement and the desire and concentration of the pupils [49,50].

POMS is commonly used to evaluate the mood state changes in participants during physical activity [50]. Several authors have reported that the state of mood varies according to the modalities of physical activity [40]. These authors indicated that positive changes in mood state in participants are generally associated with motivational physical activities. The present investigation indicated a significant increase in fatigue scores in SSG-J and SSG-NJ, suggesting that both types of games induced a similar subjective feeling of tiredness. The score of the POMS variables was positively altered with the presence of jokers. The tension score (25%) was significantly decreased, while vigor scores was increased by 30%. The significant decrement in tension scores and the increment in positive mood states (vigor) of POMS in OCs lead to a decrease in TMD scores. This indicates that games with jokers improve mood states. 

Aydi et al. [33] indicated that vigor plays a crucial role in positive change of mood, and Selmi et al. [40] suggested that the vigor that follows motivational physical activities causes positive mood, energy increment and physical engagement. We believe that motivational factors may explain the improvement in some of the tested mood’s subscales. The motivation is linked to the integration of the OCs in the games and the obligation of their partners to pass the ball to them at least once every 10 successive passes. These findings are consistent with those obtained by Sahli et al. [38] who reported that the 4-a-side SSG modality with PE teachers’ verbal encouragement results in a significant decrease in tension and TMD scores, and a significant improvement in positive mood states (vigor). Along the same lines, Aydi et al. [33] claimed that lack of motivation in physical activity leads to mood state improvement with lower fatigue and tension scores and higher vigor. According to our findings, OCs’ participation in SSG-NJ leads to an inability to concentrate and a lack of physical engagement because their normal weight partners ignore them and do not give them many passes during the game. In this case, the OCs felt neglected by their partners. Therefore, the absence of motivation in this type of exercise does not lead to the improvement of mood and physical enjoyment. These results suggest that, unlike motivational exercises, lack of encouragement during SSG-NJ causes negative feelings in OCs. 

The present study indicates that the role of the PE teacher in integrating OCs into activities and exercises during PE sessions may influence their behavior and exercise intensity and influence psychophysiological and affective responses. We can therefore say that the positive status of the OCs and their integration into the PE session and the group is associated with motivation, encouragement and a positive teaching style. However, several limitations must be considered when interpreting the present results. First, the study sample was small due to the difficulty of recruiting many participants because the number of obese children in study classes taught by the same teacher is small. Secondly, this study was conducted with male OCs under the age of 14. Therefore, it is of interest to conduct further investigations with other groups to see the physiological and affective responses during exercise for different sex and ages. Thirdly, it can be checked whether the variation in the number of ball passes, the pitch size and even the use of goalkeepers can provide additional information. Finally, the present study was carried out during PE sessions and not during a long-term learning program (e.g., learning cycle).

This investigation was performed in a real training area with OCs, thus providing useful suggestions. To our knowledge, this study is the first to examine psychophysiological aspects, physical enjoyment, and mood state during an SSG among OC students. The integration of OCs into the group by their participation as jokers is a very effective method. The joker role can be considered an essential factor during PE sessions as it induces significant physiological solicitations and positive affective responses. 

## 5. Conclusions

The present investigation shows that HRmax, RPE, positive mood and PACES scores are higher during the SSG-J in comparison with the SSG-NJ. Moreover, as SSGs with and without jokers affect many players’ performance parameters, it may be concluded that introducing these games into training could have positive feedback on OCs motivation, physical commitment and willingness to play. Thus, practitioners should apply this knowledge in their professional settings when using team games during PE sessions, as the role of the joker is an effective factor in improving internal intensity, physical enjoyment, physiological responses and mood states in OCs. Additionally, researchers can be expected to conduct surveys with larger sample groups in fields of different sizes and with diverse player numbers for further research on this topic.

## Figures and Tables

**Figure 1 children-10-00133-f001:**
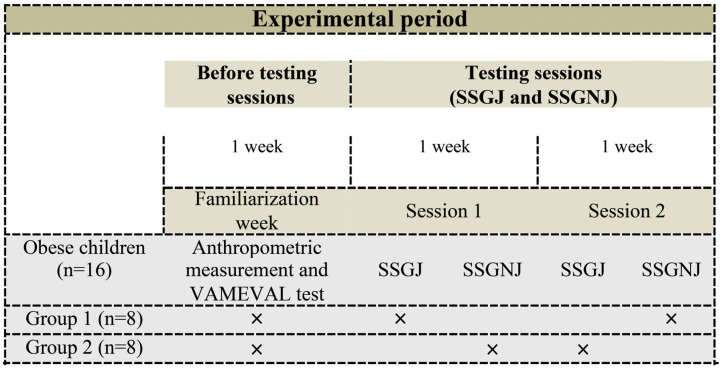
Representative diagram of the experimental protocol. SSG-J, SSG with joker; SSG-NJ, SSG without joker.

**Figure 2 children-10-00133-f002:**
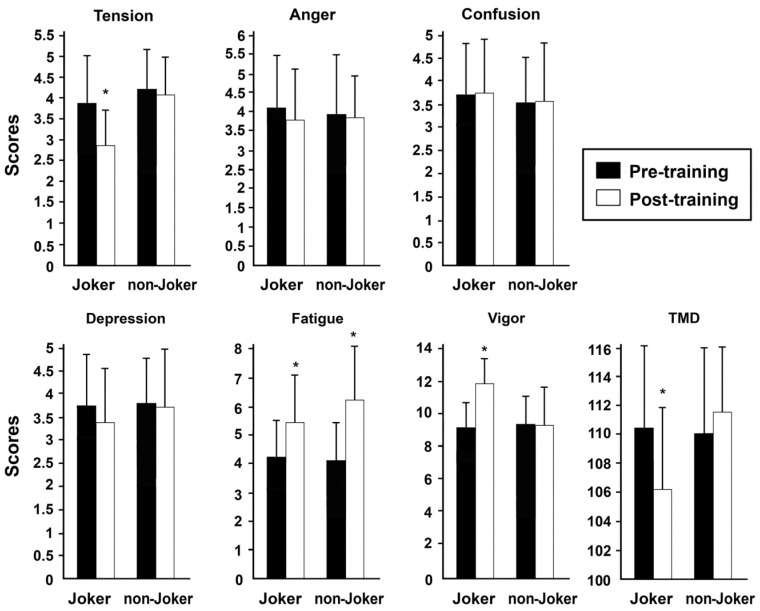
POMS scores measured before and after small-sided games (SSG) with or without joker. * *p* < 0.001, by ANOVA.

**Table 1 children-10-00133-t001:** The anthropometric characteristic of the participating children.

Measures	Mean ± S.D.
Age (years)	13.8 ± 0.73
Height (cm)	159.93 ± 3.28
Body mass (kg)	79.06 ± 3.47
BMI (kg/m^2^)	30.90 ± 0.86
%Fat	30.9 ± 0.86

BMI, body mass index; %Fat, body fat percentage; Mean ± S.D, Mean ± standard deviation.

**Table 2 children-10-00133-t002:** Comparison of measures of exercise intensity between joker and non-joker OCs.

	Joker	Non-Joker	*p* Value
HR-max (bpm)	84.06 ± 2.26	78.94 ± 3.02	<0.0001
Lactate	5.19 ± 1.22	3.94 ± 0.85	<0.01
RPE	6.5 ± 1.15	5.44 ± 0.96	<0.01

**Table 3 children-10-00133-t003:** POMS scores measured before and after small-sided games (SSG) with or without joker.

Variables	Main Effect of the Condition		Main Effect of the Times		Interaction Effect	
	F (1, 15)	η^2^	F (1, 15)	η^2^	F (1, 15)	η^2^
Tension	5.417 *	0.09	1.154	0.01	5.4 *	0.105
Anger	0.5556	0.006	0.06276	<0.0001	0.07246	<0.0001
Confusion	0.04478	<0.001	<0.0001	<0.0001	0.2195	0.004
Depression	0.2113	0.002	0.05535	<0.0001	0.2113	0.002
Fatigue	38.68 ***	0.262	32.11 ***	0.309	1.154	0.005
Vigor	51.86 ***	0.458	0.007	<0.0001	12.13 **	0.31
TMD	4.8 *	0.019	2.138	0.011	9.421 **	0.054

TMD: total mood disturbance. Significant effects are shown, with * *p* < 0.05, ** *p* < 0.01, *** *p* < 0.001.

## Data Availability

Not applicable.

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
