# Peer review of "Integration of Obese Children in Physical Education Sessions: An Exploratory Study"

_children, 2023, doi:10.3390/children10010133_

Round 1

Reviewer 1 Report

Dear authors, I reviewed your article - Integration of obese children in physical education sessions- which aim was to examine the impact of the of the OC's position in the role of Joker on psychophysiological aspects, physical enjoyment and mood state while completing a SSG exercises (Games of 10 successive passes) in PE sessions. 

I have some suggestions:

1. In table 1 describe the abbreviations used.

2. At Material Method 2.4 and 2.9 should be one subchapter because approach same topic.

3. In Material method you do not mentioned which statistics was used (appear in Results ANOVA and Mann-Whitney test).

4. If I understand well the study last 16 weeks? If the answer is YES, at Results I recommend to present in different tables pre and post results for HR max, anthropometric , OMNI, POMS and PACES data with or without joker.

5. In table 2 for P value use sign < in each situation.

6. Lines 386 and 387 must be eliminated.

7. References 23 and 25 are too old, use especially references form last 5 years, maybe 10 years ago. I suggest to remove those two references and to find another more recent.

8. Reference 40 present a study where subjects have between 18 and 30 years, your study average age is 13.8 years old. I recommend to exclude this one nd to find another more appropriate.

Author Response

I would like to thank you for the editorial report on our manuscript.

We have made appropriate amendments to the manuscript following the suggestions of the referees and we provide below a point-to-point answers to the referee’s questions.

Hoping that the revised MS will be considered suitable for publication, I look forward to your editorial reply.

                                                                       Yours Sincerely

                                                                       Prof. A. Muscella

Point-to-point reply to the Referee’s comments

Reviewer 1

In table 1 describe the abbreviations used.

The abbreviations used were described.

At Material Method 2.4 and 2.9 should be one subchapter because approach same topic.

Thank you for the suggestion. The two subchapters have been merged

In Material method you do not mentioned which statistics was used (appear in Results ANOVA and Mann-Whitney test).

A subchapter Statistical Analysis was added

If I understand well the study last 16 weeks? If the answer is YES, at Results I recommend to present in different tables pre and post results for HR max, anthropometric, OMNI, POMS and PACES data with or without joker.

As schematized in figure 1, the study lasts three weeks. Each children performed 2 SSG sessions (one SSG session with joker and one SSG session without joker); physiological and psychological measurements were performed each experimental session.

In table 2 for P value use sign < in each situation.

We have done

Lines 386 and 387 must be eliminated.

Lines 386 and 387 were eliminated

References 23 and 25 are too old, use especially references form last 5 years, maybe 10 years ago. I suggest to remove those two references and to find another more recent.

Some references have been replaced and many added.

Reference 40 present a study where subjects have between 18 and 30 years, your study average age is 13.8 years old. I recommend to exclude this one  to find another more appropriate.

The reference 40 has also been replaced

Reviewer 2 Report

December 5, 2022

Manuscript: Integration of obese children in physical education sessions

Journal: Children

First Review

OVERVIEW:

I thank you for submitting work to this journal. Clearly the author or authors are veteran writers, and their skill is observable through their work.

Not to be taken as an insult, but a compliment, this paper reads extremely simply and straightforward. It is not easy to write in a way that is so straightforward the work appears simple. Again, understand this is a complement of the writing style and structure.

As my research methodological skills are more in line with qualitative and interpretive work, I may not be an adequate reviewer regarding the merit and rigor of the quantitative data analysis. I trust other reviewers to provide a thorough point of you on the data analysis and interpretation.

What follows are my comments and suggestions to hopefully improve the clarity and presentation of this work. Please recognize these are mere suggestions from a place of intended helpfulness.

1. ADEQUACY OF LITERATURE REVIEW:

The literature review is not bad, but it is very short. Certainly, much more could be said about the current status of health, physical activity, and initiative to combat obesity since 2020. I highly recommend supplementing the literature review with more evidence to support your rationale. 

2. THEORY TESTING:

To that end, there are also health behavior theories which interact with many concepts of obesity, PE curriculum models, and sport pedagogy. Connecting the TGFU/Sport Education model to your proposed explanations would strengthen the manuscript. Connect [and please, define] the Joker role to other positions of the sport – this is necessary.

3. KNOWLEDGE GAINED:

I have concerns that your quantitative data does provide a broad descriptive overview of your inquiry, but little meaning is made from it because your observed group is so small. N=16 with no control group is not adequate for point making. Consider terming this study as an 'exploratory inquiry' or 'pilot', because a N=16 will not qualify.

4. METHODOLOGY:

I complement the author on their methodological approach. It appears sound; but a GPower analysis will prove that the number of participants needs to be multiplied by several and a control group must be included with do not receive an intervention.

5. ANALYSES:

Again, I refer the author to my comments on my research expertise being in the areas of interpretive analysis. My assumption is data analysis was done appropriately, but I would ask the authors of the work to consider commentary provided by a quantitative data expert, rather than myself.

6. INTERPRETATION:

Regarding interpretation, I have no concerns that the author has interpreted the data adequately inappropriately with the glaring concern of N=16.

7. LIMITATIONS:

I complement the author for pointing out particular limitations in their work. It gives the work trustworthiness.

8. FUTURE DIRECTIONS/APPLICATIONS:

I recommend concluding the paper with a stronger emphasis on how the work can be applied, and what future directions a line of inquiry can take in this area. The author makes an attempt at. I recommend the author extend this attempt to include specific examples of such questions and research extension that could follow. For example, leave the reader who is a researcher how and what one might do to extend this line of inquiry. Leave the reader who is a practitioner knowing how to apply this learned knowledge in their professional setting.

Author Response

I would like to thank you for the editorial report on our manuscript.

We have made appropriate amendments to the manuscript following the suggestions of the referees and we provide below a point-to-point answers to the referee’s questions.

Hoping that the revised MS will be considered suitable for publication, I look forward to your editorial reply.

                                                                       Yours Sincerely

                                                           Antonella Muscella

Reviewer 2

I thank you for submitting work to this journal. Clearly the author or authors are veteran writers, and their skill is observable through their work.

 Not to be taken as an insult, but a compliment, this paper reads extremely simply and straightforward. It is not easy to write in a way that is so straightforward the work appears simple. Again, understand this is a complement of the writing style and structure.

 As my research methodological skills are more in line with qualitative and interpretive work, I may not be an adequate reviewer regarding the merit and rigor of the quantitative data analysis. I trust other reviewers to provide a thorough point of you on the data analysis and

Thanks so much for the positive feedback

  1. ADEQUACY OF LITERATURE REVIEW:

The literature review is not bad, but it is very short. Certainly, much more could be said about the current status of health, physical activity, and initiative to combat obesity since 2020. I highly recommend supplementing the literature review with more evidence to support your rationale. 

According to the indications of the referee, we have integrated the introduction:

Obesity in children has become a major health problem all over the world. Recent estimates indicated that Over 340 million children and adolescents aged 5–19 years were overweight or obese in 2020 [1]. These states associated with a raised risk of physical and physiological diseases as well as of serious psychological consequences [2]. Thus, childhood obesity is a challenge for health systems worldwide and many international organizations are putting strategies and plans in place to decrease its prevalence [3]. Today's children lead an increasingly sedentary lifestyle that implies the time spent by playing video games, using computers/smartphones, and watching television [4]. This lifestyle leads them to neglect the physical activity (PA) typical of this period of development [5]; this has negative implications in that PA has several positive effects on the growth of children and teens during puberty.

In September 2020, the WHO Regional Committee for Europe put a new European strategy of Work for the period 2020–2025, that focuses on united actions for improving health, by promoting physical activity and healthy nutrition to contrast obesity [6]. PA in children and adolescents benefits a range of medical conditions, including cardiovascular disease, obesity and a range of psycho-logical health problems [7]. However, usually, children with obesity (OC) find a lot of difficulty in PA sessions [8]. In fact, research indicates that obesity leads to fatigue, tension, low self-esteem, poor body image, and exercise avoidance [9]. In addition, in school OC are often stigmatized as being unmotivated, less competent, lacking in self-discipline or less athletic to participate in physical education (PE) [8, 10].

  1. THEORY TESTING:

To that end, there are also health behavior theories which interact with many concepts of obesity, PE curriculum models, and sport pedagogy. Connecting the TGFU/Sport Education model to your proposed explanations would strengthen the manuscript. Connect [and please, define] the Joker role to other positions of the sport – this is necessary.

Thanks to the suggestions of the referee, we have added:

The PE teacher must create inclusive and safe learning climates for OC in PE classes or to provide different situations (i.e., methods, exercises) in terms of their unique characteristics [15]. Mostly, the need to differentiate teaching to meet the needs of students obliges PE teacher chooses methods and styles of learning to the needs of schoolchildren [18]. One of the more actual pedagogical models in the field of PE to introduce students in the sports training is the one indicated as Teaching Games for Understanding (TGfU). The TGfU pedagogical approach improves decision making, enjoyment, psychological aspects, and students’ intention to be physically active [19, 20].

Some studies indicated the effectiveness of TGfU to gain OC motivation, engagement, enjoyment, and cognitive learning [21, 22]. Generally, the programming of specific and adapted exercises for OC in PE sessions resulted in positive changes in physical engagement, student motivation, with positive effects on affective aspects [23, 13]. Physical activities help children and adolescents to decrease anxiety, improve mood, and evolve their mental health [24, 25], thus stimulant PE sessions could be an effective strategy to integrate OC and promote positive physical engagement and communication with other students [13]. In addition, the research of the motivation and enjoyment in the PE class has a great interest due to the relation between intrinsic motivation, enjoyment, and intention of being physically active for students [26, 27].

In addition, motivational exercises in PE sessions have been shown to modulate the emotional and physiological states of students which may contribute to its beneficial effect on the intensity of effort, positive feeling, and higher physical enjoyment [28]. On the other hand, non-motivating running exercises negatively affect mood state and pleasure in OC [29,30]. The role of the PE teacher in the motivation and integration of OCs in the group effectively contributes to the development of more positive physical, behavioral, and emotional states in these OCs, which give them greater confidence and integration into the group [16, 17].

 As the motivating exercises are fun for the students, they can encourage the OC to become more active [31].

And also:

The joker is used to assist other players because it is involved in both the defensive and offensive phases of the game. More, the role of joker player constrains the game by forcing the defender player to adapt to a new game context [41]. Using joker players is a common practice during training sessions [41, 42, 43], however, there are no investigations regarding OC integration.

Using the joker position during ball games in PE sessions could be important and beneficial for OC, because it is more effective for learning and motivating than other methods.

  1. KNOWLEDGE GAINED:

I have concerns that your quantitative data does provide a broad descriptive overview of your inquiry, but little meaning is made from it because your observed group is so small. N=16 with no control group is not adequate for point making. Consider terming this study as an 'exploratory inquiry' or 'pilot', because a N=16 will not qualify.

Thank you for the suggestion. We modified the title.

  1. METHODOLOGY:

I complement the author on their methodological approach. It appears sound; but a GPower analysis will prove that the number of participants needs to be multiplied by several and a control group must be included with do not receive an intervention.

In this research, we investigated the effect of the role of the joker in children with obesity (OCs) on integration, physio-psychological responses during Small-Sided Games training programs. We then compared the parameters measured in children with obesity in the joker role with those obtained measured in non-joker roles. Therefore, from this point of view, the control is represented by the SSG-NJ group.

  1. FUTURE DIRECTIONS/APPLICATIONS:

I recommend concluding the paper with a stronger emphasis on how the work can be applied, and what future directions a line of inquiry can take in this area. The author makes an attempt at. I recommend the author extend this attempt to include specific examples of such questions and research extension that could follow. For example, leave the reader who is a researcher how and what one might do to extend this line of inquiry. Leave the reader who is a practitioner knowing how to apply this learned knowledge in their professional setting.

Thanks to the suggestions of the referee, we have added:

The present investigation shows that HRmax, RPE, positive mood and PACES score are higher during the SSG-J in comparison with SSG-NJ. Furthermore, as SSGs with and without jokers affect numerous performance parameters of the players, it can be concluded that the implementation of these games to training can have a positive effect on OCs motivation, physical commitment, and willingness to play. Thus, practitioners should apply this knowledge in their professional setting when using team games during PE sessions, since the role of the joker is an effective factor in improving internal intensity, physical enjoyment, physiological responses, and mood state in OCs. Besides, it may be suggested that researchers should conduct studies with larger sample groups in different field sizes and with different number of players for further research on this subject.

Reviewer 3 Report

The idea of the study is interesting, my recommendations are the following:

In abstract

I recommend that you mention in detail what PACES, POMS represent.

Lines 17-18 I recommend you also mention the duration/period when the training program was implemented.

For the results, I recommend that you mention numerically the functional results obtained.

Keywords - detail what they represent: SSG

Lines 89-92 recommend rewriting because it sounds like a conclusion.

I recommend expanding the Introduction section by mentioning relevant bibliographic indexes. Index 14 appears 7 times, it is too much.

According to the sample from the abstract, the children were soccer players, and the Small-sided games session is based on hand passes, made with the handball. In this context, the usefulness of the selected program does not emerge. I recommend you to clarify this aspect.

Also in section 2.4 you mention 3 times, bibliographic index 14, without being relevant, I recommend clarification.

The OMNI-Child perceived exertion scales- is not mentioned in the abstract as an evaluation tool, I recommend the correction.

Line 212 mentioned - a 5-point Likert scale (0 means "Not at all" and 4 means "Extremely", I recommend clarification.

Lines 216-217 recommend that you clearly mention Cronbach's α for the entire questionnaire and each subscale.

Lines 220-221 mention the bibliographic index that refers to adults and not to children. I recommend that you explain how this questionnaire was adapted to the age of the subjects of this study.

Section 2.8 bibliographic index 15 is not relevant. I recommend clarification.

Section 2.9 re repeat and in section 2.5 with the exception of blood lactate concentration. I recommend the correction.

  The Statistical Analyzes section is missing from the article, I recommend the introduction.

I recommend deleting fig.2, it duplicates the information from the previous sentence.

I recommend that in Table 3 you insert a few columns in which to enter the test results before and after the program. What is the arithmetic mean and standard deviation for each item in the two tests. I recommend deleting fig. 3 and making clarifications.

I recommend that the Discussions section be organized into subsections. I also recommend that such a study mention at least 70 bibliographic indexes. In this sense, I recommend the expansion of the discussion section with the realization of concrete correlations between the present results and results from previous studies.

Lines 386-387 recommend deletion.

In conclusion, I consider that the implementation of a program aimed at a motor skill specific to handball does not fit with the selection of subjects. If it refers only to children who practice physical education, then this study is relevant. I recommend a rethinking and clarification.

Author Response

Reviewer 3

In abstract

I recommend that you mention in detail what PACES, POMS represent.

We have done.

Lines 17-18 I recommend you also mention the duration/period when the training program was implemented.

We have done.

For the results, I recommend that you mention numerically the functional results obtained.

According to the referee’s suggestions, we revised the abstract.

Keywords - detail what they represent: SSG

We have done.

Lines 89-92 recommend rewriting because it sounds like a conclusion.

We have rewriting:

Using the joker position during ball games in PE sessions could be important and beneficial for OC, because it is more effective for learning and motivating than other methods. Furthermore, this specific intervention could provide physiological and psychological benefits due to positive sensations and high effort

I recommend expanding the Introduction section by mentioning relevant bibliographic indexes. Index 14 appears 7 times, it is too much.

According to the indications of the referee, we have integrated the introduction, some references have been replaced and many added.

According to the sample from the abstract, the children were soccer players, and the Small-sided games session is based on hand passes, made with the handball. In this context, the usefulness of the selected program does not emerge. I recommend you to clarify this aspect.

It was an oversight. The abstract has been revised

Also in section 2.4 you mention 3 times, bibliographic index 14, without being relevant, I recommend clarification.

We have corrected

The OMNI-Child perceived exertion scales- is not mentioned in the abstract as an evaluation tool, I recommend the correction.

We have corrected

Line 212 mentioned - a 5-point Likert scale (0 means "Not at all" and 4 means "Extremely", I recommend clarification.

We have done

Lines 216-217 recommend that you clearly mention Cronbach's α for the entire questionnaire and each subscale.

We have done

Lines 220-221 mention the bibliographic index that refers to adults and not to children. I recommend that you explain how this questionnaire was adapted to the age of the subjects of this study.

We have changed the reference

Section 2.8 bibliographic index 15 is not relevant. I recommend clarification.

We have deleted bibliographic index 15

Section 2.9 re repeat and in section 2.5 with the exception of blood lactate concentration. I recommend the correction.

The two subchapters have been merged

The Statistical Analyzes section is missing from the article, I recommend the introduction.

A subchapter Statistical Analysis was added

I recommend deleting fig.2, it duplicates the information from the previous sentence.

We have deleted fig.2.

I recommend that in Table 3 you insert a few columns in which to enter the test results before and after the program. What is the arithmetic mean and standard deviation for each item in the two tests. I recommend deleting fig. 3 and making clarifications.

We thank the referee for the suggestion, but we think that using graphical format to visually represent the relationship between two types of data (SSG-J vs SSG-NJ and pre- vs post-training) could be more understandable, even without the need to read the text.

I recommend that the Discussions section be organized into subsections. I also recommend that such a study mention at least 70 bibliographic indexes. In this sense, I recommend the expansion of the discussion section with the realization of concrete correlations between the present results and results from previous studies.

We have integrated the introduction and organized into subsections.

Lines 386-387 recommend deletion.

We have done

In conclusion, I consider that the implementation of a program aimed at a motor skill specific to handball does not fit with the selection of subjects. If it refers only to children who practice physical education, then this study is relevant. I recommend a rethinking and clarification.

We would like to clarify (as indicated in the title) that the intervention consisted of physical education sessions, aimed at integrating children with obesity.

Round 2

Reviewer 2 Report

I am very impressed with the authors' reaction to reviewer feedback - the revision addresses every concern I had from the original submission. I believe the manuscript is greatly improved by virtue of a more robust literature review, more current citation, and greater clarity to the variables observed, measured, and reported upon. 

This work is scientific, relevant, and adequately presented. Very well done.

Author Response

We thank the reviewer for the positive feedback.

Reviewer 3 Report

The article review is adequate with the recommendations.

From an aesthetic point of view, I recommend the following:

Lines 373-382 these discussions recommend to be grouped in a single sentence, because they refer to the same idea.

Line 383 - to delete the space between the lines.

Author Response

Thankyou for suggestions.
Lines 373-382 are grouped and clarified.
The space between the lines is deleted.